# BMP7 Increases UCP1-Dependent and Independent Thermogenesis with a Unique Gene Expression Program in Human Neck Area Derived Adipocytes

**DOI:** 10.3390/ph14111078

**Published:** 2021-10-25

**Authors:** Abhirup Shaw, Beáta B. Tóth, Rini Arianti, István Csomós, Szilárd Póliska, Attila Vámos, Zsolt Bacso, Ferenc Győry, László Fésüs, Endre Kristóf

**Affiliations:** 1Laboratory of Cell Biochemistry, Department of Biochemistry and Molecular Biology, Faculty of Medicine, University of Debrecen, H-4032 Debrecen, Hungary; abhirup.shaw@med.unideb.hu (A.S.); toth.beata@med.unideb.hu (B.B.T.); ariantirini@med.unideb.hu (R.A.); vamos.attila@med.unideb.hu (A.V.); 2Doctoral School of Molecular Cell and Immune Biology, University of Debrecen, H-4032 Debrecen, Hungary; 3Department of Biophysics and Cell Biology, Faculty of Medicine, University of Debrecen, H-4032 Debrecen, Hungary; csomos.istvan@med.unideb.hu (I.C.); bacso@med.unideb.hu (Z.B.); 4Genomic Medicine and Bioinformatics Core Facility, Department of Biochemistry and Molecular Biology, Faculty of Medicine, University of Debrecen, H-4032 Debrecen, Hungary; poliska@med.unideb.hu; 5Faculty of Pharmacy, University of Debrecen, H-4032 Debrecen, Hungary; 6Department of Surgery, Faculty of Medicine, University of Debrecen, H-4032 Debrecen, Hungary; gyory@med.unideb.hu

**Keywords:** BMP7, adipocyte, UCP1, thermogenesis, creatine cycle, obesity

## Abstract

White adipocytes contribute to energy storage, accumulating lipid droplets, whereas brown and beige adipocytes mainly function in dissipating energy as heat primarily via the action of uncoupling protein 1 (UCP1). Bone morphogenic protein 7 (BMP7) was shown to drive brown adipocyte differentiation in murine interscapular adipose tissue. Here, we performed global RNA-sequencing and functional assays on adipocytes obtained from subcutaneous (SC) and deep-neck (DN) depots of human neck and differentiated with or without BMP7. We found that BMP7 did not influence differentiation but upregulated browning markers, including UCP1 mRNA and protein in SC and DN derived adipocytes. BMP7 also enhanced mitochondrial DNA content, levels of oxidative phosphorylation complex subunits, along with PGC1α and p-CREB upregulation, and fragmentation of mitochondria. Furthermore, both UCP1-dependent proton leak and UCP1-independent, creatine-driven substrate cycle coupled thermogenesis were augmented upon BMP7 addition. The gene expression analysis also shed light on the possible role of genes unrelated to thermogenesis thus far, including ACAN, CRYAB, and ID1, which were among the highest upregulated ones by BMP7 treatment in both types of adipocytes. Together, our study shows that BMP7 strongly upregulates thermogenesis in human neck area derived adipocytes, along with genes, which might have a supporting role in energy expenditure.

## 1. Introduction

Two functionally distinct types of fat are present in mammals: white and brown adipose tissue. Healthy human adults possess thermogenic brown adipose tissue (BAT) that can dissipate energy as heat under sub-thermal conditions [1,2]. These depots are primarily located in six anatomical regions: supraclavicular, axillary, mediastinal, cervical, paravertebral, and abdominal [3,4,5]. Supraclavicular, paravertebral, and deep-neck (DN) regions contain the highest amounts of thermogenic adipose tissue. Together, these depots account for almost 5% of the basal metabolic rate, thereby playing an important role in combating weight gain and type 2 diabetes [6,7]. The thermogenic adipocytes in rodents are either classical brown or beige cells distinguished by their origin and distribution [8,9,10,11]. Non-shivering thermogenesis of interscapular BAT and inguinal white adipose tissue (WAT) depots of rodents is majorly mediated by uncoupling protein 1 (UCP1), which is mitochondria resident and functions via generating proton leak in the inner mitochondrial membrane thereby uncoupling the mitochondrial respiratory chain from ATP synthesis [8,12,13,14]. A UCP1-independent mechanism for thermogenesis has been recently discovered in murine beige fat, which is mediated via the futile creatine phosphate cycle [15,16,17]. In this process, creatine kinase first phosphorylates creatine using ATP. Creatine phosphate is immediately dephosphorylated, most probably by tissue-nonspecific alkaline phosphatase (TNAP), generating heat [18]. Novel strategies for specific stimulation of beneficial fat burning by the metabolically active BAT in humans can aid weight reduction and decrease insulin resistance in individuals with obesity [6,7].

Bone morphogenic proteins (BMPs) act via heterotetrametric complexes of transmembrane receptors, which are type I or type II serine/threonine kinase receptors [19,20,21]. Type I BMP receptors (BMPR) are divided into BMPR1A, BMPR1B, Activin A Receptor (ACVR)-like (L) 1, and ACVR1. BMP7 has been known to interact with BMPR1A, BMPR1B, and ACVR1. It also interacts with three distinct type II receptors: BMPR2, ACVR2A, and ACVR2B [22,23,24]. BMP7 acts as an auto/paracrine mediator that drives classical brown adipocyte differentiation in mice; it promotes differentiation of brown preadipocytes even in the absence of an adipogenic induction cocktail [25]. BMP7 increases the expression of UCP1, peroxisome proliferator-activated receptor-gamma coactivator (PGC)1α, and PRDM16 via the activation of p38 mitogen activated protein (MAP) kinase pathway [25]. BMP7 has also been shown to upregulate UCP1 expression in selected clones of immortalized white and brown preadipocytes derived from human neck [26]. Further studies are necessary to better understand the molecular mechanisms by which BMP7 can exert its thermogenic effects on distinct types of human adipocytes.

Recently, we screened and compared global gene expression patterns by RNA-sequencing of human primary differentiated adipocytes derived from progenitors of DN and subcutaneous neck (SC) origins [27,28]. Consistent with previous studies [26,29,30,31,32], DN derived adipocytes displayed higher browning features than the SC derived ones. The purpose of the present study was to learn whether BMP7 can enhance thermogenic differentiation of human neck area derived adipocytes. Our results indicate that BMP7 caused a significant upregulation of UCP1 protein expression in both SC and DN adipocytes that was further confirmed by immunostaining. BMP7 treatment enhanced mitochondrial biogenesis, fragmentation, and UCP1-dependent oxygen consumption rate (OCR). We also detected an increase in creatine driven substrate cycle coupled OCR and induction of genes related to this futile cycle, which generates heat without UCP1 [15,18,33]. The global gene expression data revealed pathways thus far unrelated to thermogenesis but upregulated as a result of BMP7 administration.

## 2. Results

### 2.1. SC and DN Derived Preadipocytes Are Differentiated to Adipocytes to the Same Extent in the Absence or Presence of BMP7

Adipose tissue biopsies of SC and DN origins were obtained in pairs from nine independent donors, followed by isolation and cultivation of preadipocytes according to previously described protocols [27]. White adipogenic differentiation medium was applied to the preadipocytes for 14 days, with or without BMP7. A heatmap prepared from data of global RNA-sequencing for illustrating the expression of general adipocyte marker genes (e.g., LEP, FABP4, ADIPOQ, PPARG) showed similarly elevated expression in BMP7 treated and untreated differentiating adipocytes (Figure 1A). Laser-scanning cytometry based quantification of adipogenic differentiation rate showed that approximately 60% of the cells were differentiated upon administration of the white adipogenic differentiation medium; the presence of BMP7 did not influence the differentiation rate (Figure 1B) [34,35]. Gene expression of BMPR subunits, such as BMPR1A, BMPR1B, ACVR1, BMPR2, ACVR2A, and ACVR2B were abundant at the preadipocyte stage in both SC and DN derived adipocytes, their expression was not affected by BMP7 significantly (Figure 1C–H).

### 2.2. BMP7 Upregulated Browning Marker Genes Including UCP1 in SC and DN Derived Differentiated Adipocytes

A recent in-depth analysis of white, brown, and beige adipocyte transcriptomes of murine and human origins has been utilized to develop an algorithm named, BATLAS that can effectively calculate brown/beige adipocyte content in cell and biopsy samples [36]. BATLAS analysis of gene expression data from the nine independent donor samples revealed that brown/beige content of DN samples was significantly higher than of SC ones, which was further elevated upon BMP7 treatment (Figure 2A). The presence of BMP7 also increased the browning capacity of SC derived adipocytes. Next, texture sum variance was quantified using laser-scanning cytometry [34,35], which revealed that BMP7 treatment slightly decreased the size of lipid droplets in both SC and DN derived adipocytes, that suggests increased capacity for lipolysis and energy expenditure (Figure 2B). Gene expression of UCP1 tended to elevate upon BMP7 treatment in both types of adipocytes as determined by RNA-sequencing (Figure 2C). This upregulation was further confirmed by RT-qPCR analysis and was found to be statistically significant in the case of DN adipocytes (Figure 2D). Consistent with previous results, DN derived adipocytes expressed more UCP1 protein than SC derived ones, which was significantly upregulated in response to BMP7 in adipocytes of both origins (Figure 2E). Additionally, immunostaining clearly revealed an increase of the UCP1 intensity upon BMP7 treatment in adipocytes differentiated from both types of progenitors (Figure 2F; see secondary antibody control in Appendix A). Laser-scanning cytometry assisted quantification of UCP1 immunostaining intensity showed a significant increase in DN adipocytes as a result of BMP7 (Figure 2G) [34,35]. Together these data indicate that BMP7 upregulates browning marker genes, including UCP1, in primary human neck area derived adipocytes.

### 2.3. BMP7 Facilitated Mitochondrial Biogenesis Leading to Increased Protein Expression of Mitochondrial OXPHOS Complex Subunits

Next, we addressed the question how BMP7 treatment influences mitochondrial biogenesis and content. The protein expression of PGC1α, one of the principal regulators of mitochondrial biogenesis [37,38], was found to be significantly upregulated upon BMP7 treatment in both SC and DN derived adipocytes (Figure 3A). PGC1α can also be induced by the transcription factor CREB, which can bind to a functional CRE in the PGC-1 promoter [37]. A significant induction of CREB phosphorylation in DN adipocytes further suggested an increase in mitochondrial biogenesis upon BMP7 treatment via upregulation of CREB pathway (Figure 3B). The amount of mitochondrial respiratory chain complex subunits was also elevated upon BMP7 treatment. Complex I, II, and III subunits showed significant upregulation upon BMP7 treatment in both types of adipocytes, while Complex IV was significantly upregulated in case of DN adipocytes. Complex V showed an increasing trend, although statistically not significant, in both SC and DN derived adipocytes (Figure 3C). Collectively, these data clearly demonstrate that BMP7 upregulates mitochondrial biogenesis in SC and DN derived adipocytes.

### 2.4. BMP7 Treatment Increased Mitochondrial Fragmentation and Elevated Cellular Respiration with Enhanced Proton Leak

UCP1-enriched mitochondria are mostly fragmented [39]. Immunostaining of translocase of outer mitochondrial membrane 20 (TOM20) was performed to visualize mitochondrial morphology. The content of fragmented mitochondria was significantly higher in differentiated DN derived adipocytes compared to SC derived ones, which corresponds to their higher UCP1 content and thermogenic capacity. BMP7 significantly increased the fragmented mitochondria content in both SC and DN derived adipocytes (Figure 4A,B; see Appendix A, for secondary antibody control). Total mitochondrial DNA content was also increased significantly upon BMP7 treatment in both SC and DN derived adipocytes (Figure 4C). These data suggest that mitochondria of adipocytes differentiated in the presence of BMP7 possess an enhanced thermogenic potential.

Next, we intended to investigate the functional consequences of the increased mitochondrial content and fragmentation. Monitoring of OCR revealed that both basal and 3′,5′-cyclic adenosine monophosphate (cAMP) stimulated OCR was significantly elevated upon BMP7 treatment in SC and DN derived adipocytes. Importantly, proton leak related OCR, revealed by oligomycin addition, was significantly elevated, which indicates increased thermogenesis mediated by UCP1 (Figure 5A). Basal extracellular acidification rate (ECAR) showed an increasing trend upon BMP7 treatment, while cAMP stimulated ECAR was significantly elevated in DN derived adipocytes differentiated in the presence of BMP7 (Figure 5B).

### 2.5. Creatine Driven Substrate Cycle Related Thermogenesis Was Facilitated upon BMP7 Treatment

Recent studies in mice have shown that a futile creatine phosphate cycle plays an important role in thermogenic metabolism of beige adipocytes [15]. Our research group demonstrated the contribution of this cycle to thermogenesis in ex vivo models of human subcutaneous and neck area adipocytes [40,41,42]. We could observe significant increase of creatine driven substrate cycle related OCR, revealed by applying the creatine transport inhibitor, β-guanidinopropionic acid (β-GPA) [11], in both SC and DN derived adipocytes differentiated in the presence of BMP7 (Figure 6A). RNA expression of mitochondrial creatine kinase 2 (CKMT2), one of the possible kinases acting in the futile cycle, was found slightly elevated in the generated RNA-sequencing dataset, and this was confirmed by RT-qPCR analysis in SC and DN adipocyte samples upon BMP7 treatment. Importantly, the CKMT2 protein level was also significantly higher in these adipocytes (Figure 6B). A recent publication has shown that cytosolic creatine kinase B (CKB) is targeted to mitochondria and is indispensable for thermogenesis by the creatine phosphate cycle [33]. Evaluation of RNA-sequencing data revealed an increasing trend for CKB expression upon BMP7 treatment. BMP7 caused significant upregulation of CKB protein expression in DN derived adipocytes (Figure 6C). The phosphatase TNAP has been identified to hydrolyze phosphocreatine to creatine, thereby driving the futile creatine phosphate cycle in mitochondria of thermogenic adipocytes [18]. TNAP was found to be expressed at a high extent in both SC and DN derived adipocytes, however, its gene expression remained unchanged upon BMP7 treatment (Figure 6D). Together, these data suggest that BMP7 upregulates thermogenesis not only via UCP1 but also by the futile creatine phosphate cycle in human neck area adipocytes.

### 2.6. BMP7 Upregulates Genes, Including Aggrecan (ACAN), Crystallin Alpha B (CRYAB), and Inhibitor of Differentiation 1 (ID1), Thus Far Not Linked to Thermogenesis

Next, we aimed to explore whether BMP7 treatment could upregulate pathways in neck area derived adipocytes, which have not been related to thermogenesis thus far. Further analysis of RNA-sequencing data revealed that 121 and 60 genes were upregulated in SC and DN derived adipocytes in response to BMP7 treatment, while 190 (such as COL6A6, MMP27, MGAT4C, CCL11, and LINC01028) and 87 (such as KRT1, MYH8, CCL11, MATN4, and FAM180B) genes were downregulated, respectively. Genes, up- or downregulated by BMP7, were visualized by Volcano Plot (Figure 7A) and clustered by a heatmap (Appendix A). The lists of all the genes upregulated (Appendix A) and downregulated (Appendix A) are provided in descending order of their fold change. 38 genes were commonly upregulated between SC and DN derived adipocytes, while 45 genes were commonly downregulated, respectively (Figure 7B). Gephi illustration of Panther pathway analysis illustrates that signaling by BMP involving SMAD group of transcription factors (SMAD6, SMAD7, SMAD9) was commonly upregulated in both SC and DN adipocytes (Table 1, Appendix A), which is consistent with previously published results [43]. Integrin cell surface interactions, involving ITGA9, COMP, or ITGA8 and GPCR ligand binding, involving ADRA2A, ADRA2C, FZD1, FZD5, or ACKR1 were among the significantly elevated pathways only in SC derived adipocytes upon BMP7 treatment (Table 1, Appendix A). Extracellular matrix proteins, involving ACAN were also found to be upregulated only in SC derived adipocytes (Table 1).

According to the RNA-sequencing analysis, ACAN, CRYAB, and ID1 were strongly upregulated upon BMP7 treatment in both types of adipocytes (Figure 7A). Gene expression of ACAN was confirmed to be significantly higher in SC adipocytes upon BMP7 treatment by RT-qPCR analysis, protein expression of ACAN also followed a similar pattern (Figure 7C). Gene expression of CRYAB, a member of small heat shock protein (HSP) 20 family, was found to be elevated upon BMP7 treatment by RNA-sequencing and the upregulation was further confirmed by RT-qPCR in both SC and DN adipocytes. CRYAB protein expression was similarly elevated (Figure 7D). ID1 gene expression was also significantly higher in SC and DN derived adipocytes differentiated in the presence of BMP7 as compared to untreated ones quantified by both RNA-sequencing and RT-qPCR analysis. ID1 protein expression was significantly elevated in SC derived adipocytes, while in DN adipocytes it showed a trend of upregulation (Figure 7E).

## 3. Discussion

BMP7 functions as an autocrine/paracrine mediator that promotes classical brown and beige adipocyte differentiation in mice [25,44]. Pre-treatment with BMP7 of immortalized human neck derived white and brown preadipocyte clones, followed by the administration of an adipogenic induction medium, also significantly increased UCP1 gene expression [26]. Our research group observed that BMP7 can significantly upregulate gene expression of UCP1 and the classical brown adipocyte marker, ZIC1 in differentiated Simpson–Golabi–Behmel syndrome (SGBS) adipocytes [42]. BMP7 has also shown to improve insulin signal transduction in cultured human hepatocytes [45]. BMP7 exerts its effect via upregulation of p38 MAP kinase and PGC1α, which leads to upregulation of mitochondrial biogenesis and UCP1 expression [25]. BMP7 can also act via phosphorylation of SMAD 1/5/8 [43].

In our study, the receptors involved in BMP signaling [19,20,22,23] were found to be abundantly expressed in the preadipocyte stage irrespective of the anatomical origin of the progenitors isolated from human neck area (Figure 1). BMP7 significantly upregulated UCP1-dependent thermogenesis when applied on top of a white adipogenic differentiation medium in human primary SC and DN derived adipocytes (Figure 2). BMP7 treatment led to increased protein expression of mitochondrial oxidative phosphorylation complex subunits (Figure 3C) and total mitochondrial DNA content (Figure 4C) in parallel with the upregulation of PGC1α and p-CREB pathways (Figure 3A,B), which is consistent with previously published studies [25]. Our study first showed that BMP7 elevated fragmented mitochondria content in both SC and DN derived adipocytes (Figure 4A,B), which indicates an increased thermogenic potential [39]. This was further proved by a significant increase in basal, stimulated, and proton leak OCR in both types of adipocytes differentiated in the presence of BMP7 (Figure 5). 

Kazak et al. discovered that a creatine-mediated futile cycle is responsible for enhanced mitochondrial respiration in beige fat [15]. A recent publication from the same group showed that an adipose tissue specific knockout of glycine amidinotransferase, the rate limiting enzyme for creatine biosynthesis, made mice prone to diet-induced obesity [46]. This clearly illustrated the importance of creatine metabolism in energy expenditure in vivo. Another study in mice showed that creatine driven thermogenesis plays an important role in both UCP1 positive and negative beige adipocytes [47]. Mitochondrial creatine kinases, CKMT1 A/B, CKMT2, and most recently CKB have been postulated to play an important role in the UCP1-independent thermogenesis mediated by the futile cycle [15,33]. Previously, our research group has shown that BMP7 treatment in SGBS adipocytes significantly elevated creatine cycle related OCR [42]. We also demonstrated that clozapine-induced human browning adipocytes of abdominal origin increased their energy expenditure via the futile creatine cycle [41]. Our previous study showed that mitochondrial creatine kinases, CKMT1 A/B were expressed at a greater extent in DN compared to SC adipocytes; creatine cycle related OCR followed a similar pattern as well [27]. We report for the first time that BMP7 treatment significantly increases creatine phosphate cycle related thermogenesis, which was evident from a strong increase in creatine cycle related OCR (Figure 6A). We also observed an increased gene and protein expression of mitochondrial CKMT2 (Figure 6B) and CKB (Figure 6C), which further supports this conclusion.

It is well characterized that long-term administration of rosiglitazone induces browning of human adipocytes ex vivo [27,34,48] and increases beiging of WAT in mice [49]. In SC adipocytes, only seven genes (AF131216.3, TMEM132C, SLC7A10, STOX1, RIMS4, PLPR4, and CMYA5) were commonly upregulated and ten (TMEM204, TMEM158, EPHB6, PLA2G5, RSPO1, DPT, TMC2, GPM6B, TRPV6, and HPSE) were downregulated, respectively, by both BMP7 and rosiglitazone [27]. Out of the commonly upregulated ones, SLC7A10 encodes the alanine-serine-cysteine transporter-1 (ASC-1), which also showed higher expression in DN derived adipocytes as compared to SC derived ones [40]. Expression of SLC7A10 in human WAT inversely correlates with visceral obesity and insulin resistance [50]. ASC-1 promoted mitochondrial respiration of adipocytes, especially when they were stimulated for thermogenesis by a cell permeable cAMP analogue [40,50]. In DN adipocytes, only RSPO3 was commonly upregulated and nine genes (WISP1, CLSTN2, LTBP2, DPT, IFITM1, INMT, TMEM158, TNC, and RSPO1) were downregulated, respectively, by both treatments [27]. Our presented results have also revealed certain genes, such as ACAN, CRYAB, or ID1, unrelated to human adipocyte browning thus far, which show a strong correlation with increased thermogenesis upon BMP7 treatment suggesting that they might have a positive effect on thermogenesis. Further research is necessary to unravel their mechanism of action and molecular targets with respect to thermogenesis.

Aggrecan protein, encoded by ACAN, is a chondroitin sulphated proteoglycan, which functions as a critical structural component of cartilage [51]. Aggrecan is also found in the brain almost exclusively in the perineuronal net and is postulated to play an important role in its formation and function [52,53,54,55] and a similar task might be also needed in the innervation of browning adipose tissue. Gene and protein expression of ACAN was significantly upregulated in BMP7 treated SC derived adipocytes (Figure 7C).

CRYAB is one of the major structural proteins of eye lens and is found to be expressed in several other tissues. It belongs to the small HSP family and behaves as a chaperone, thereby protecting against apoptosis and oxidative stress [56,57,58,59,60,61]. In the lens, CRYAB is a substrate for transglutaminase 2 (TGM2)-mediated cross-linking [62,63]. Our group found that gonadal WAT of TGM2^−/−^ mice exhibited a lower expression of beige marker genes, such as UCP1, TBX1, and TNFRFS9; therefore, these mice were cold intolerant [64]. CRYAB has also been identified as a novel adipokine and its protein expression is strongly induced during adipogenesis, reaching a 10-times higher level in mature human white adipocytes than in preadipocytes [65,66]. Gene expression of CRYAB was higher in visceral WAT of children with obesity as compared to normal-weight [67]. CRYAB protein expression was found to be higher in subcutaneous WAT of old as compared to young individuals with obesity [68]. CRYAB mRNA had high expression in thermogenically more potent DN as compared to SC adipocytes and was less expressed in adipocytes that carried the FTO obesity-risk genotype [27]. In addition, gene and protein expression of CRYAB increased significantly upon BMP7 treatment in SC and DN derived adipocytes (Figure 7D) in parallel to increased thermogenic potential pointing to a thus far not recognized significance of CRYAB in this process.

The ID proteins (ID 1–4) are a subfamily of helix–loop–helix (HLH) transcription factors lacking a basic DNA binding domain. ID proteins act predominantly via dimerization with other transcriptional regulators, mostly by basic-HLH (bHLH) factors, which fail to bind to DNA and thereby function as dominant negative regulators of bHLH proteins [69]. A recent study that screened protein expression patterns of different metabolically active tissues showed that ID1 is highly expressed in both WAT and BAT of adult mice, with BAT showing the highest expression [70]. However, the same study found that an adipose tissue specific overexpression of ID1 resulted in high fat diet-induced obesity in mice. Depletion of ID1 increased UCP1 gene expression in mouse WAT upon cold exposure [71]. Another study indicated that ID1 deficiency improves glucose tolerance and lowers serum insulin levels in mouse WAT [72,73]. Our study here shows that BMP7 significantly upregulated ID1 gene and protein expression in SC and DN derived adipocytes (Figure 7E). The increased ID1 expression correlated with increased UCP1 expression, mitochondrial biogenesis, and content. Functional experiments are required in the future to explore the direct effect of ID1 on human adipocyte thermogenesis.

## 4. Materials and Methods

### 4.1. Materials

All chemicals were obtained from Sigma Aldrich (Munich, Germany) unless otherwise stated.

### 4.2. Isolation, Maintenance, Differentiation, and Treatment of Human Neck Derived Human Adipose-Derived Stromal Cells (hASCs)

hASCs were isolated from the stromal-vascular fractions of SC and DN tissues of patients (35–75 years) undergoing a planned elective surgery. Biopsies were obtained in pairs from SC and DN regions to avoid inter-individual variations. Patients with known diabetes, abnormal thyroid hormone levels or malignant tumors were not included in this study. Written informed consent was obtained from all participants prior surgery [27,28,74,75].

hASCs were isolated, maintained, and checked for mycoplasma contamination as previously described [27,74]. Cells were grown to 90% confluency and differentiated following a reported white adipogenic differentiation protocol, with or without BMP7 (R&D Systems, Minneapolis, MN, USA, 354-BP) at 50 ng/mL [27,34]. Media was replaced at an interval of 4 days, followed by collection of cells on day 14. 

### 4.3. RNA Isolation, RT-qPCR, and RNA-Sequencing

Cells were collected in Trizol reagent (Thermo Fisher Scientific, Waltham, MA, USA), followed by manual RNA isolation by chloroform extraction and isopropanol precipitation. Global transcriptome data were obtained via high throughput mRNA-sequencing [27]. Pathways were obtained using Panther Reactome Pathways (http://www.pantherdb.org/, access date 20 September 2021). Heatmaps were constructed on the Morpheus web tool (https://software.broadinstitute.org/morpheus, access date 20 September 2021) with Pearson correlation of rows and complete linkage based on calculated z-score of DESeq normalized data after log2 transformation [27]. Gephi images were constructed as previously described [27]. Browning content was calculated using the BATLAS Webtool (http://green-l-12.ethz.ch:3838//BATLAS, access date 20 September 2021) [36]. 

RNA quality was evaluated by Nanodrop (Thermo Fisher Scientific), followed by generation of cDNA by TaqMan reverse transcription reagent kit (Thermo Fisher Scientific) and qPCR analysis [40]. Normalized gene expression was quantified using the probes listed in Appendix A).

### 4.4. Western Blot, Immunoblotting and Antibodies

Samples were separated by SDS-PAGE, transferred to PVDF membrane, and blocked by 5% skimmed milk solution as previously described [40,42,76]. The following primary antibodies were used overnight in 1% skimmed milk solution: anti-UCP1 (1:750, R&D Systems, MAB6158), anti-pCREB (1:1000, Merck-Millipore, Burlington, MA, USA, 05-667), anti-CREB (1:1000, Abcam, Cambridge, UK, ab31387), anti-PGC1α (1:1000, Santa Cruz Biotechnology, Dallas, TX, USA, H-300), anti-CKMT2 (1:1000, Novus Biologicals, Centennial, CO, USA, NBP2-13841), anti-CKB (1:1000, Novus Biologicals, NBP1-84460), anti-CRYAB (1:1000, Novus Biologicals, NB100-2519), anti-β actin (1:5000, Novus Biologicals, A2066), anti-OXPHOS (1:1000, Abcam, ab110411), anti-Aggrecan (1:1000, Novus Biologicals, NB100-74350), and anti-ID1 (1:1000, Novus Biologicals, JM92-13). HRP-conjugated goat anti-rabbit (1:10,000, Advansta, San Jose, CA, USA, R-05072-500) or anti-mouse (1:5000, Advansta, R-05071-500) IgG were used as secondary antibodies, respectively. Immunoreactive proteins were visualized by Immobilion Western chemiluminescence substrate (Merck-Millipore). Densitometry was carried out by FIJI.

### 4.5. Immunostaining and Image Analysis

SC and DN derived hASCs were plated and differentiated in 8-well Ibidi μ-chambers (Ibidi GmbH, Gräfelfing, Germany). Cells were washed with PBS prior to fixation by 4% paraformaldehyde. Cells were permeabilized with 0.1% saponin, followed by blocking with 5% skimmed milk as per previously described protocols [76]. The cells were incubated overnight with the following primary antibodies: anti-TOM20 (1:75, WH0009804M1) or anti-UCP1 (1:200, U6382) at room temperature. Incubation with the following secondary antibodies was kept for 3 h at room temperature: Alexa 647 goat anti-mouse IgG (1:1000, Thermo Fischer Scientific, A21236) or Alexa 488 goat anti-rabbit IgG (1:1000, Thermo Fischer Scientific, A11034). Nuclei were labeled with Propidium Iodide (1.5 μg/mL, 1 h). Images were obtained by Olympus FluoView 1000 confocal microscope. Fragmented mitochondria were defined by the size (pixel^2^) 0–100 AU and counted using FIJI(ImageJ) [76]. Adipogenic differentiation rate, texture sum variance, and UCP1 immunostaining intensity were quantified using laser-scanning cytometry as previously described [34,35].

### 4.6. Determination of Cellular OCR and ECAR

OCR and ECAR were measured by XF96 oximeter (Seahorse Biosciences, North Billerica, MA, USA). Cells were seeded and differentiated on XF96 assay plates with or without BMP7. During measurement, baseline respiration and acidification were measured for 30 min followed by stimulated OCR and ECAR. A single bolus dose of dibutyryl-cAMP (at 500 μM final concentration) was added to mimic adrenergic stimulation leading to the stimulated OCR and ECAR, which were measured at 30-min intervals. 3 h post treatment, oligomycin at 2 μM final concentration was added to block ATP synthase activity to measure proton leak OCR. For measuring creatine cycle related OCR, β-GPA was added at 2 mM concentration after cAMP treatment. Antimycin A at 10 μM final concentration was added at the end for baseline correction. The OCR was normalized to protein content [40].

### 4.7. Statistics and Image Preparation

Results are expressed as mean ± SD for the number of independent repetitions indicated. One-way ANOVA with Tukey’s post hoc test were used for multiple comparisons of groups. Two-tailed paired t test was used to compare between two individual groups. Graphpad Prism 9 was used to visualize the graphs and evaluation of statistics.

## 5. Conclusions

As a conclusion, to the best of our knowledge, we showed for the first time that BMP7 can significantly facilitate thermogenesis in human primary SC and DN derived ex vivo differentiating adipocytes. The increased thermogenesis was mediated primarily via UCP1 dependent proton leak and, to some extent, by the futile creatine phosphate substrate cycle. We demonstrated that both fragmented mitochondria content and mitochondrial OXPHOS complex subunits were elevated upon BMP7 treatment. The global RNA-sequencing analysis gave clue to novel genes, such as ID1, ACAN, and CRYAB, which might play a supportive role in thermogenesis. Further functional experiments are necessary to validate and confirm their possible action with this regard.

## Figures and Tables

**Figure 1 pharmaceuticals-14-01078-f001:**
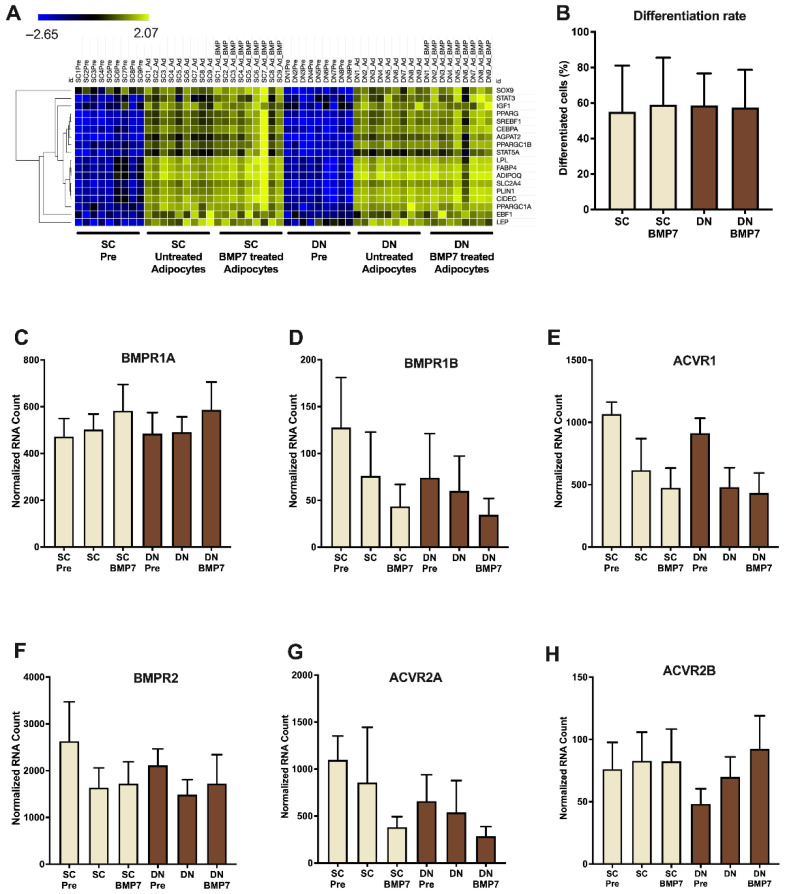
Preadipocytes derived from subcutaneous (SC) and deep-neck (DN) depots of human neck widely expressed Bone morphogenic protein (BMP) 7 receptors and differentiated equally in the presence or absence of BMP7. SC and DN derived preadipocytes (Pre) were differentiated for two weeks following the white adipogenic differentiation protocol, BMP7 (50 ng/mL) was added where indicated. (**A**) Heatmap showing the expression pattern of general adipogenic differentiation marker genes and (**B**) quantification of differentiation rate assisted by laser-scanning cytometry. Quantification of gene expression of (**C**) BMPR1A, (**D**) BMPR1B, (**E**) ACVR1, (**F**) BMPR2, (**G**) ACVR2A, and (**H**) ACVR2B by RNA-sequencing. Data expressed as mean ± SD. *n* = 9.

**Figure 2 pharmaceuticals-14-01078-f002:**
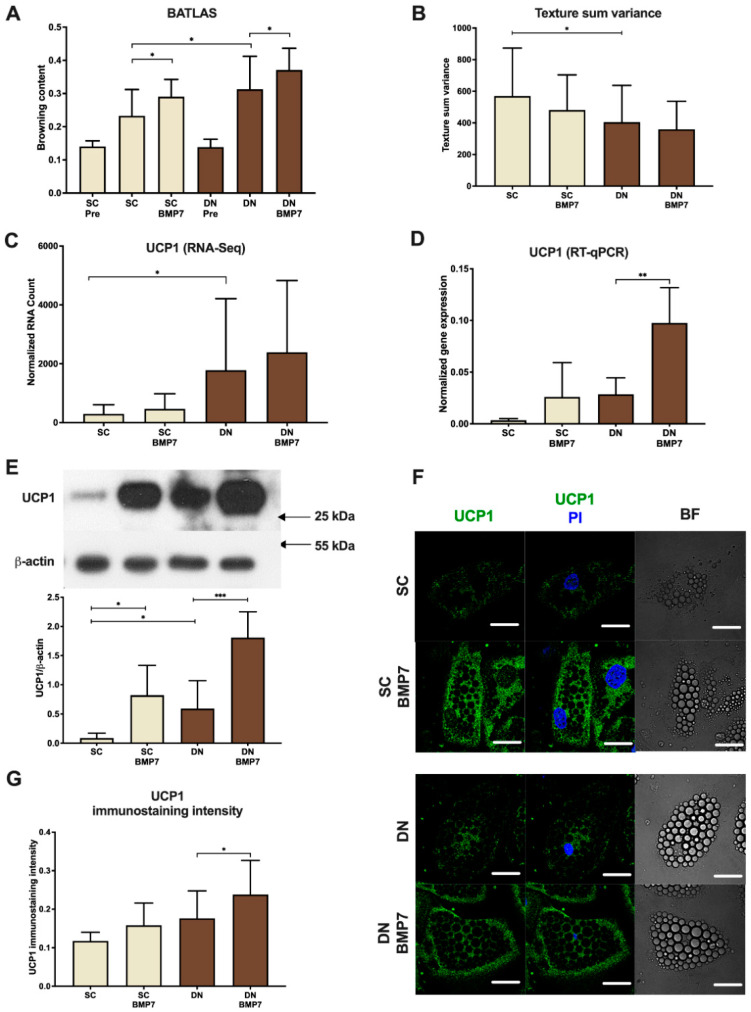
BMP7 upregulated browning markers in both SC and DN derived differentiated adipocytes. SC and DN preadipocytes were differentiated and treated as in Figure 1. (**A**) Browning content as estimated by BATLAS Webtool (*n* = 9); (**B**) texture sum variance as quantified by laser-scanning cytometry (*n* = 5); quantification of UCP1 gene expression by (**C**) RNA-sequencing (*n* = 9) and (**D**) RT-qPCR normalized to GAPDH (*n* = 5). (**E**) UCP1 protein expression normalized to β-actin (*n* = 5); (**F**) representative UCP1 immunostaining images visualized by confocal microscopy. Nucleus was stained using propidium iodide (PI), BF represents bright field image. Scale bars represent 10 μm. (**G**) Quantification of UCP1 immunostaining intensity assisted by laser-scanning cytometry (*n* = 4000 cells of 4 donors). Data expressed as mean ± SD. * *p* < 0.05, ** *p* < 0.01, *** *p* < 0.001. Statistics: Paired *t*-test (**A**,**B**,**G**), GLM (**C**), and one-way ANOVA with Tukey’s post-test (**D**,**E**).

**Figure 3 pharmaceuticals-14-01078-f003:**
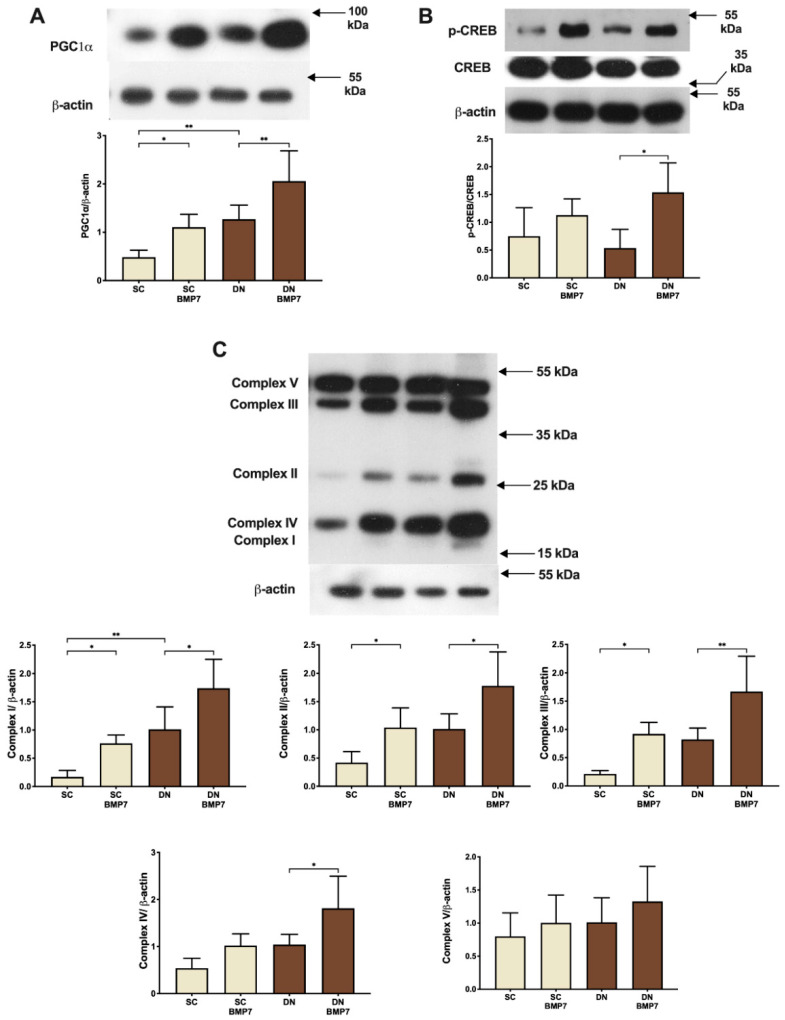
BMP7 treatment increased mitochondrial biogenesis and protein expression of mitochondrial OXPHOS complex subunits in SC and DN derived differentiated adipocytes. SC and DN preadipocytes were differentiated and treated as in Figure 1 and Figure 2. (**A**) Protein expression of PGC1α normalized to β-actin, (**B**) Ratio of p-CREB to CREB protein expression, and (**C**) protein expression of mitochondrial OXPHOS complexes (I, II, III, IV, and V) normalized to β-actin. Data expressed as mean ± SD. *n* = 5. * *p* < 0.05, ** *p* < 0.01. Statistics: one-way ANOVA with Tukey’s post-test.

**Figure 4 pharmaceuticals-14-01078-f004:**
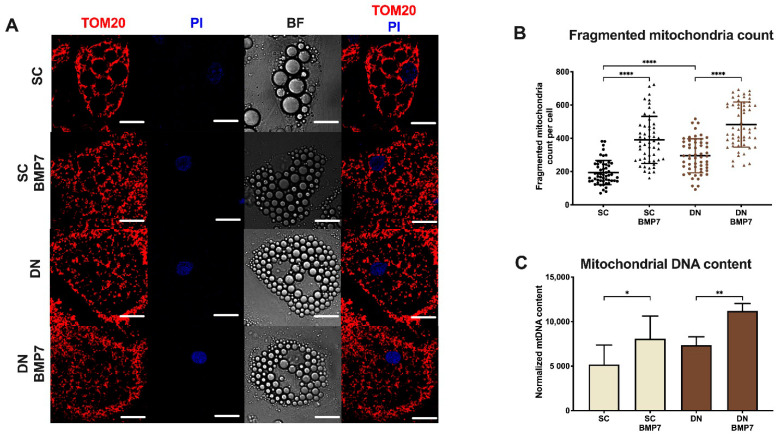
Fragmented and total mitochondrial content were increased upon BMP7 treatment in SC and DN derived differentiated adipocytes. SC and DN preadipocytes were differentiated and treated as in Figure 1, Figure 2 and Figure 3. (**A**) Representative confocal microscopy images of TOM20 immunostaining. PI was used to stain the nucleus. Scale bars represent 10 μm. (**B**) Quantification of fragmented mitochondrial content normalized to per nuclei (*n* = 55 cells of 3 donors); (**C**) total mitochondrial DNA content quantified by qPCR (*n* = 5). Data expressed as mean ± SD. * *p* < 0.05, ** *p* < 0.01, **** *p* < 0.0001. Statistics: one-way ANOVA with Tukey’s post-test.

**Figure 5 pharmaceuticals-14-01078-f005:**
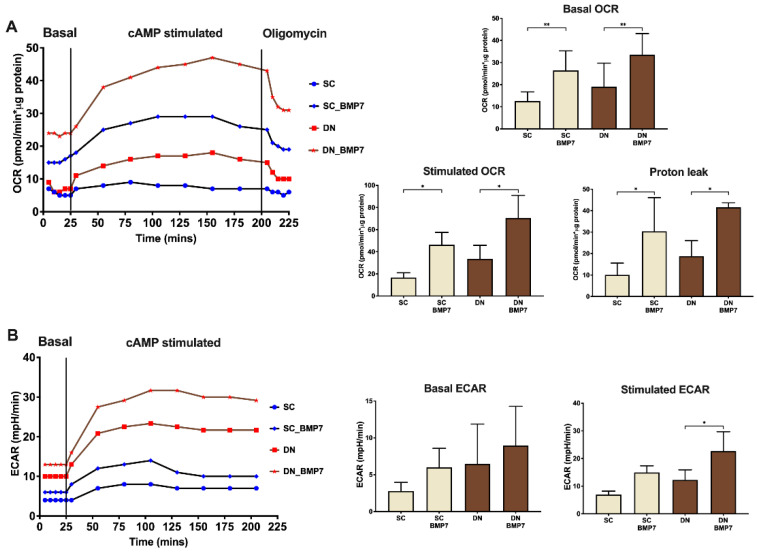
BMP7 treatment resulted in increased cellular respiration, thermogenic response, and extracellular acidification in SC and DN derived differentiated adipocytes. SC and DN preadipocytes were differentiated and treated as in Figure 1, Figure 2, Figure 3 and Figure 4. (**A**) Representative time lapse curve for measured oxygen consumption rate (OCR) after indicated treatments, followed by quantification of basal, stimulated, and proton leak OCR; (**B**) representative time lapse curve for measured extracellular acidification rate (ECAR) after the indicated treatment, followed by quantification of basal and stimulated ECAR. Data expressed as mean ± SD. *n* = 4. * *p* < 0.05, ** *p* < 0.01. Statistics: one-way ANOVA with Tukey’s post-test.

**Figure 6 pharmaceuticals-14-01078-f006:**
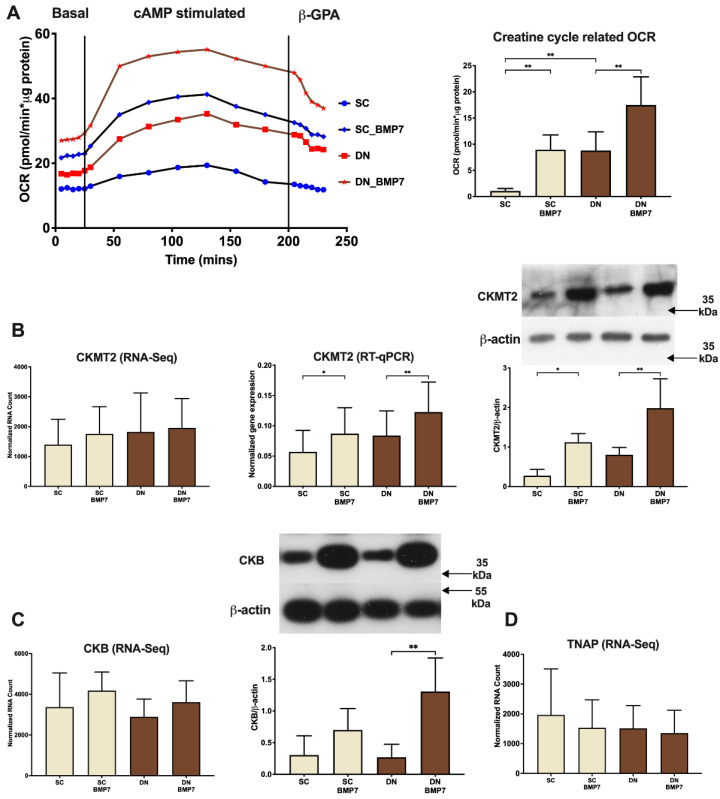
BMP7 treatment upregulated creatine futile cycle-mediated thermogenesis in SC and DN derived differentiated adipocytes. SC and DN preadipocytes were differentiated and treated as in Figure 1, Figure 2, Figure 3, Figure 4 and Figure 5. (**A**) Representative time lapse OCR curve after indicated treatments, followed by quantification of creatine cycle related (β-GPA inhibited) OCR (*n* = 5). (**B**) Quantification of gene expression of CKMT2 by RNA-sequencing (*n* = 9) and RT-qPCR (*n* = 5), followed by protein expression (*n* = 5). (**C**) Quantification of gene expression of CKB by RNA-sequencing (*n* = 9), followed by protein expression (*n* = 4). (**D**) Quantification of gene expression of TNAP by RNA-sequencing (*n* = 9). Data expressed as mean ± SD. RT-qPCR was normalized to GAPDH, protein expression was normalized to β-actin. * *p* < 0.05, ** *p* < 0.01. Statistics: one-way ANOVA with Tukey’s post-test.

**Figure 7 pharmaceuticals-14-01078-f007:**
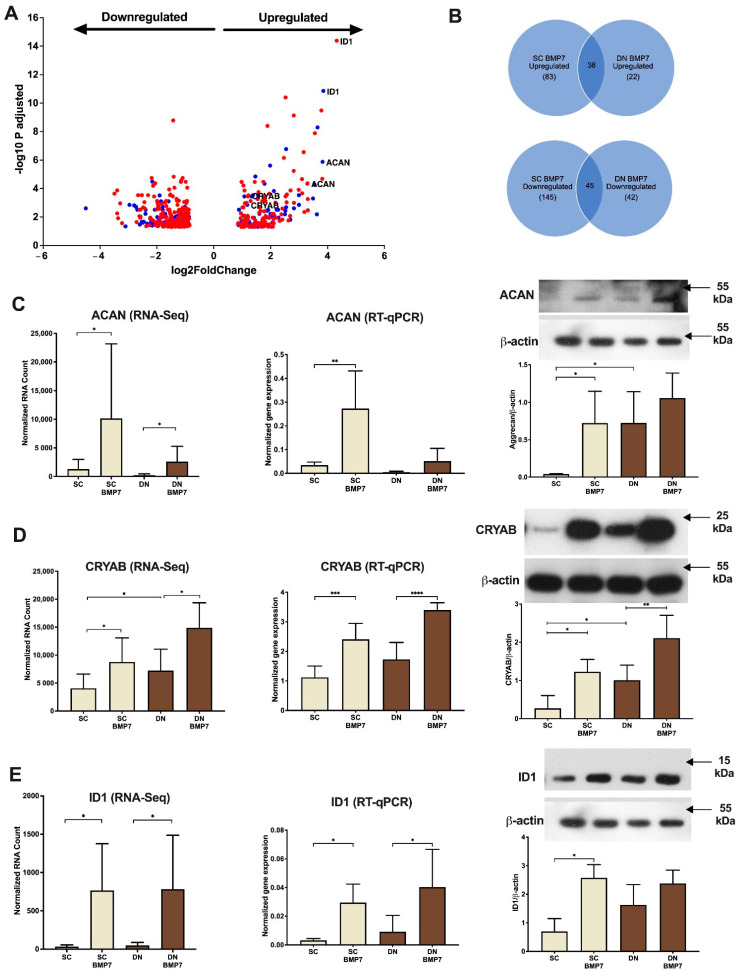
Genes upregulated upon BMP7 treatment in SC and DN derived differentiated adipocytes. SC and DN preadipocytes were differentiated and treated as in Figure 1, Figure 2, Figure 3, Figure 4, Figure 5 and Figure 6. (**A**) Volcano plot illustrating each of the upregulated genes in SC (red) and DN (blue) depots upon BMP7 treatment. (**B**) Venn-diagram showing the genes commonly up- or downregulated by BMP7 in SC and DN derived adipocytes. Gene expression of (**C**) ACAN, (**D**) CRYAB, and (**E**) ID1 as determined by RNA-sequencing (*n* = 9) and RT-qPCR (*n* = 5), followed by their respective protein expression (*n* = 5). Data presented as mean ± SD. RT-qPCR was normalized to GAPDH, protein expression was normalized to β-actin. * *p* < 0.05, ** *p* < 0.01, *** *p* < 0.001, **** *p* < 0.0001. Statistics: GLM (RNA-sequencing), one-way ANOVA with Tukey’s test (RT-qPCR and densitometry of western blots).

**Table 1 pharmaceuticals-14-01078-t001:** Pathways of genes significantly upregulated upon BMP7 treatment in SC and DN derived differentiated adipocytes. Genes commonly upregulated in SC and DN derived adipocytes are marked red. False discovery rate (FDR) values are indicated.

**SC BMP7 Upregulated**		
**Reactome Pathways**	**Gene Name**	**FDR**
**Signaling by BMP**	NOG, SMAD9, SMAD6, SMAD7	2.49 × 10^−2^
**ECM proteoglycans**	ITGA9, AL645608, ITGA8, ACAN, COMP, COL9A3, COL9A2	1.87 × 10^−3^
**Integrin cell surface interactions**	ITGA9, AL645608, ITGA8, COMP, COL9A3, COL9A2	2.3 × 10^−2^
**GPCR ligand binding**	PLPPR4, NMUR1, ADRA2C, FZD5, ACKR1, ADRA2A, PTH1R, NPY1R, CHRM4, FZD1, EDNRA, PTGDR, CNR1	2.93 × 10^−2^
**DN BMP7 Upregulated**		
**Reactome Pathways**	**Gene Name**	**FDR**
**Signaling by BMP**	NOG, SMAD9, SMAD6, SMAD7	6.05 × 10^−3^

## Data Availability

RNA-sequencing data were deposited to Sequence Read Archive (SRA) database [https://www.ncbi.nlm.nih.gov/sra] (access date 30 September 2021) under accession number PRJNA607438. Other data that support the findings of this study are available from the corresponding authors (fesus@med.unideb.hu, kristof.endre@med.unideb.hu) upon reasonable request.

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
