# Peer review of "BMP7 Increases UCP1-Dependent and Independent Thermogenesis with a Unique Gene Expression Program in Human Neck Area Derived Adipocytes"

_pharmaceuticals, 2021, doi:10.3390/ph14111078_

Round 1

Reviewer 1 Report

The authors describe the effects of bone morphogenic protein (BMP7) on differentiation and markers of browning in both subcutaneous and deep-neck (DN) adipocyte depots. They found that BMP7 did not impact differentiation but did elevate UCP-1 mRNA and protein. In addition, it also increased mitochondrial DNA content, fragmentation, and oxidative phosphorylation complex subunits as well as upregulate PGC1alpha and p-CREB. The authors also found that BMP7 also increased UCP-1 dependent and UCP-1 independent thermogenesis in both SC and DN derived adipocytes. Finally, the authors found that BMP7 upregulated ACAN, CRYAB and IDI in both types of adipocytes. They conclude that BMP7 upregulates thermogenesis and markers of thermogenic genes in human neck derived adipocytes that may play a role in energy expenditure.

Minor suggestions:

1) The authors may want to specify which adipose depot they are referring to in lines 24-27

2) The authors should point out which adipose tissue depot they are referring to when discussing UCP-1 dependent and independent mechanisms on non-shivering thermogenesis in rodent models. Are they referring to interscapular brown adipose tissue (which is the most commonly studies depot in rodents) as well as inguinal white adipose tissue for UCP-1 dependent and UCP-1 independent thermogenic mechanisms, respectively? The authors should provide more information here (lines 46-49).

3) Can the authors provide some information about the specific primary antibodies and provide evidence that the appropriate controls were also completed including replacement of antibody weight normal serum) on the adipocytes used in their study? Have their been other publications using these particular antibodies they can also refer to from their own laboratory?    

Author Response

The authors describe the effects of bone morphogenic protein (BMP7) on differentiation and markers of browning in both subcutaneous and deep-neck (DN) adipocyte depots. They found that BMP7 did not impact differentiation but did elevate UCP-1 mRNA and protein. In addition, it also increased mitochondrial DNA content, fragmentation, and oxidative phosphorylation complex subunits as well as upregulate PGC1alpha and p-CREB. The authors also found that BMP7 also increased UCP-1 dependent and UCP-1 independent thermogenesis in both SC and DN derived adipocytes. Finally, the authors found that BMP7 upregulated ACAN, CRYAB and IDI in both types of adipocytes. They conclude that BMP7 upregulates thermogenesis and markers of thermogenic genes in human neck derived adipocytes that may play a role in energy expenditure.

Answer: We thank the Reviewer for the constructive comments which helped us to improve our manuscript.

Minor suggestions:

1) The authors may want to specify which adipose depot they are referring to in lines 24-27

Answer: This information was added into the corresponding sentences of the Abstract.

2) The authors should point out which adipose tissue depot they are referring to when discussing UCP-1 dependent and independent mechanisms on non-shivering thermogenesis in rodent models. Are they referring to interscapular brown adipose tissue (which is the most commonly studies depot in rodents) as well as inguinal white adipose tissue for UCP-1 dependent and UCP-1 independent thermogenic mechanisms, respectively? The authors should provide more information here (lines 46-49).

Answer: The indicated section of the Introduction has been more detailed in the revised version of the manuscript.

3) Can the authors provide some information about the specific primary antibodies and provide evidence that the appropriate controls were also completed including replacement of antibody weight normal serum) on the adipocytes used in their study? Have their been other publications using these particular antibodies they can also refer to from their own laboratory?

Answer: In case of primary anti-UCP1 (R&D Systems, MAB6158), we have validated the specificity of the antibody by applying mouse interscapular BAT lysate as a positive control and compared to other commercially available anti-UCP1 antibodies (Ref. 42). Then, we have used this antibody in studies Ref. 27, 40, and 76. These references are now cited in 4.4 of the revised manuscript text. Anti-CKMT2 (Novus Biologicals, NBP2-13841) was applied previously in Ref. 40. Anti-OXPHOS (1:1000, Abcam, ab110411) was used previously in Ref. 40 and 76. All antibodies were obtained from suppliers that validated their specificity. As per the information provided on supplier’s website, anti-pCREB (Merck-Millipore, 05-667) has been cited by PMIDs: 23828861, 2222865367, 22442066; anti-CREB (Abcam, ab31387) has been cited by PMIDs: 33171176, 33725236, 33762011; anti-PGC1α (Santa Cruz Biotechnology, H-300) has been cited by PMIDs: 15048126, 15469941, 22711985; anti-CKB (Novus Biologicals, NBP1-84460) has been cited by PMIDs: 26460485, 24743550, 22093360; anti-CRYAB (Novus Biologicals, NB100-2519) has been cited by PMIDs: 23533150, 23395094; anti-Aggrecan (Novus Biologicals, NB100-74350) has been cited by PMIDs: 34330063, 33253804, 31972963. In all cases, a distinct band was detected at the predicted molecular weight of the protein of interest, and no other unspecific bands were found nearby. 

Reviewer 2 Report

This study shows that BMP7 strongly upregulates thermogenesis in human neck area derived adipocytes, along with genes, which might have a supporting role in energy expenditure. This topic is very interesting and the researchers have come to their own conclusion. However, more research is needed in the future.

  1. The abstract and introduction should be more clarified with current status, purpose and what are the main advanced knowledge or information.
  2. In Fig.3A, the results of WB for PGC1α seems not significant. The authors should provide a clear band.
  3. The complex V in figure 3C is not described in this paper.
  4. In figure 4A, the BF of DN group is quite different from that of the other three groups.
  5. Why is UCP1 not included in Table 1?
  6. Ensure the accuracy of the reference lists before submitting the manuscript. Some of the references cited in the paper are older studies, and should be replaced by more recent papers.

Author Response

This study shows that BMP7 strongly upregulates thermogenesis in human neck area derived adipocytes, along with genes, which might have a supporting role in energy expenditure. This topic is very interesting and the researchers have come to their own conclusion. However, more research is needed in the future.

Answer: We thank the Reviewer for the useful comments and suggestions which helped to improve the quality of our manuscript.

  1. The abstract and introduction should be more clarified with current status, purpose and what are the main advanced knowledge or information.

Answer: The abstract and introduction were amended in the revised version of the manuscript.

  1. In Fig.3A, the results of WB for PGC1α seems not significant. The authors should provide a clear band.

Answer: We have run immunoblots with adipocyte lysates of five independent donors and quantified them by densitometry. With respect to the data of five repetitions, PGC1α protein expression was increased in response to BMP7 in both SC and DN adipocytes. In the revised version, we have included another representative immunoblot for PGC1α.

  1. The complex V in figure 3C is not described in this paper.

Answer: We found an increasing trend, although statistically not significant, with regard to Complex V content in both SC and DN adipocytes. This is now included in 2.3 of the revised manuscript text.

  1. In figure 4A, the BF of DN group is quite different from that of the other three groups.

Answer: The representative immunostaining images showing DN adipocytes have been replaced with pictures having more proper visibility in the revised version of the manuscript.

  1. Why is UCP1 not included in Table 1?

Answer: Table 1 lists only those genes that were significantly upregulated in either SC or DN adipocytes as a result of BMP7 and fitted into a significantly overrepresented gene expression reactome pathway determined by Panther Pathway analysis (http://www.pantherdb.org/). Differential expression analysis was performed using DESeq2 algorithm; significantly differentially expressed genes were defined based on adjusted p values < 0.05 and log2 fold change threshold > 0.85 (Ref. 27,28). UCP1 was not amongst these genes.

  1. Ensure the accuracy of the reference lists before submitting the manuscript. Some of the references cited in the paper are older studies, and should be replaced by more recent papers.

Answer: We have updated the reference list in the revised version of the manuscript with more recent publications (Ref. 7,10,11,14,16,17,21,24,28-32) related to the field.

Round 2

Reviewer 2 Report

The manuscript has been revised accordingly.